# The Importance of 6-Aminohexanoic Acid as a Hydrophobic, Flexible Structural Element

**DOI:** 10.3390/ijms222212122

**Published:** 2021-11-09

**Authors:** Agnieszka Markowska, Adam Roman Markowski, Iwona Jarocka-Karpowicz

**Affiliations:** 1Department of Analytical Chemistry, Medical University of Bialystok, 15-089 Bialystok, Poland; iwona.jarocka-karpowicz@umb.edu.pl; 2Department of Internal Medicine and Gastroenterology, Polish Red Cross Memorial Municipal Hospital, 79 Henryk Sienkiewicz Street, 15-003 Bialystok, Poland; adromax@wp.pl

**Keywords:** 6-aminohexanoic acid, antifibrynolytics, linker

## Abstract

6-aminohexanoic acid is an ω-amino acid with a hydrophobic, flexible structure. Although the ω-amino acid in question is mainly used clinically as an antifibrinolytic drug, other applications are also interesting and important. This synthetic lysine derivative, without an α-amino group, plays a significant role in chemical synthesis of modified peptides and in the polyamide synthetic fibers (nylon) industry. It is also often used as a linker in various biologically active structures. This review concentrates on the role of 6-aminohexanoic acid in the structure of various molecules.

## 1. Introduction

6-aminohexanoic acid (ε-aminocaproic acid and 6-aminocaproic acid) is an ω-amino acid with a hydrophobic and flexible structure (Figure 1). Different abbreviations of its name are often used in scientific literature (Ahx, EACA, Aca, 6-ACA, Aha and Acp). IUPAC [1] recommends that ε-Ahx should generally be used in its shorter form: Ahx. The synthetic lysine analog, without an α-amino group, is used clinically as an antifibrinolytic drug and plays a role in the chemical synthesis of modified peptides and various biologically active structures [2].

Despite the extensive use of 6-aminohexanoic acid as a starting material in the production of nylon-6 and nylon-66 polyamides [3], the synthesis of this molecule is still based on the methods of the 1960s [4]. The methods consist in the hydrolysis of ε-caprolactam under acidic or basic conditions, and then purification with ion exchange resins. The acid is obtained in the form of a salt. Sattler et al. used six isolated enzymes to biosynthesize aminohexanoic acid from cyclohexanol [5]. The latest method of Ahx biosynthesis is the use of *Pseudomonas taiwanensis* VLB120 strains converting cyclohexane to є-caprolactone and combination with *Escherichia coli* JM101 strains ensuring further conversion to Ahx [6,7]. This combination of rationally designed strains enabled the direct production of Ahx from cyclohexane in one pot with high conversion and yield under environmentally friendly conditions.

## 2. Ahx as an Antifibrinolytic Drug

ε-Aminocaproic acid is clinically employed as an inhibitor of fibrinolysis [2,8]. Ahx is an amino acid structurally similar to the natural amino acid, is a lysine without an α-amino group, and most of its effects are probably related to this structural similarity. The potential use of Ahx as a therapeutic agent was first described in 1959 by Okamoto [9].

The site of action of Ahx is the fibrinolytic system, consisting of inactive plasminogen (Plg), Plg activators, and its active form, i.e., plasmin (Plm) (Figure 2) [10]. When plasminogen is cleaved (or “activated”) by Plg activators, it forms plasmin, the enzyme mainly responsible for fibrin proteolysis (fibrinolysis) and dissolution of clots in vivo. The kringle domains of both Plg and Plm contain “lysine binding sites” (LBS) that interact with lysine residues in other molecules, e.g., fibrin. The mechanism of action of Ahx is based on the molecules mimicking the side chain of lysine in fibrin and interacting with LBS of Plm/Plg and competitively prevents Plm/Plg from binding to fibrin(ogen) [11,12]. This action leads to the inhibition of Plm-induced fibrin degradation. Ahx inhibits the interaction of plasminogen and plasmin with fibrin ten times stronger than lysine. Lysine binding sites of plasminogen and plasmin contain doubly charged anionic (Asp55 and Asp57) and cationic centers (Arg71 and Arg34) interrupted by a hydrophobic channel (Val62 and Phe36), and Ahx “fits” perfectly in these places [10,11]. In zwitterionic ligands with antifibrinolytic activity, such as Ahx, the distance between the unsubstituted negatively charged carboxylic group and the positively charged terminal ammonium group should be close to 7 Å [12]. The conformational flexibility of the ε-aminocaproic acid is not as significant an advantage in antifibrinolytic activity as in other applications of ω-amino acid, as later described. An analog with a more rigid (cyclic) structure and a distance between the amino and carboxyl groups of 6.8 Å, i.e., tranexamic acid, shows about ten times higher antifibrinolytic activity than Ahx [13,14,15]. Although these lysine analogs are used as antifibrinolytic drugs, comparing their efficacy and safety is still an important research problem [16,17,18,19].

## 3. Amino Acid and Short Peptide Derivatives of Ahx as Antifibrynolytics

Aminocaproic α-amino acid derivatives with hydrophobic side chains also show antifibrinolytic activity. N^α^-ω-aminocaproyl-L-lysine was the first dipeptide, which shows significant antifibrinolytic activity, but still lower than Ahx. Midura-Nowaczek et al. obtained a series of derivatives (Table 1), such as HCl × H-EACA-Leu-OH (IC_50_ 0.08 mM), HCl × H-EACA-Cys (S-Bzl)-OH (IC_50_ 0.04 mM), H-EACA-L-Phe-OH (IC_50_ 0.16 mM), and–compared to EACA (IC_50_ 0.2 mM)—the most active antifibrinolytic H-EACA-NLeu-OH (IC_50_ < 0.02 mM) [20]. ε-Aminocaproyl-L-tryptophan and ε-aminocaproyl-L-tyrosine were also highly active, in contrast to compounds lacking a large amino acid side chain linked to EACA, such as HCl × H-Gly-EACA-OH or H-EACA-Ala-OH.

In research on antifibrinolytic activity, Midura-Nowaczek et al. also noted that Ahx derivatives, but not Ahx itself, show the ability to inhibit the amidolytic activity of plasmin, i.e., they act in the active center of the enzyme [20,24]. A structure–activity relationship was noticed in the obtained structures. In the case of esterification of ε-aminocaproic derivatives of glycine, leucine and S-benzylcysteine, an increase in the inhibitory activity of plasmin was observed. Replacing the branched chain with the n-alkyl chain of norleucine resulted in an improvement in the inhibitory activity. H-EACA-NLeu-EACA-OH and H-EACA-NLeu-OBzl were those compounds that turned out to be the best inhibitors of the caseinolytic activity of plasmin in this series. On the other hand, the amide derivative with the butyl group H-EACA-NLeu-NH-C_4_H_9_ turned out to be the best plasmin inhibitor in both the amidolytic and caseinolytic tests. Fifteen peptide derivatives of Ahx containing the known plasmin-affinity fragment -Ala-Phe-Lys- were synthesized and evaluated as potential plasmin inhibitors. H-D-Ala-Phe-Lys-EACA-NH_2_ was the most active and specific inhibitor of the amidolytic activity of plasmin (IC_50_ = 0.02 mM) [29].

The antiplatelet properties of the examined aminocaproic-α-amino acids are expected to reduce the antithrombotic properties of these compounds compared to Ahx. Bruzgo et al. investigated the effect of three Ahx derivatives with antifibrinolytic activity, i.e., HCl × H-EACA-L-Leu-OH, HCl × H-EACA-L-Cys(S-Bzl)-OH, and H-EACA-L-Nle-OH on platelet responses (aggregation and adhesion) and on their integrity. It was found that none of the tested compounds, up to 20 mM, were toxic to platelets [21,22]. In comparison with Ahx, all the synthetic derivatives inhibited much stronger ADP- and collagen-induced aggregation of platelets suspended in plasma (platelet-rich plasma) as well as the aggregation of these cells in whole blood. Ahx and its derivatives showed a similar inhibitory effect on the thrombin-induced adhesion of platelets to fibrinogen-coated surfaces.

Two derivatives (H-EACA-Leu-OH and H-EACA-Nle-OH) showed a cytotoxic effect against MCF-7 and fibroblast cell lines, particularly in high concentrations [30]. Six derivatives of Ahx inhibited topoisomerase II action on supercoiled DNA [31].

## 4. Introduction of Ahx into the Structure of Peptides with Biological Activity

There has been an interest for some time in developing new synthetic drugs, derived from natural peptide-based products, such as neurotoxins, antimicrobial peptides, plant peptides, and others (Table 2). Successful chemical modifications include cyclization, lipidation, glycosylation, cationization, or PEGylation. Traditional modifications of the structure are less suitable for disulfide-rich peptides, as they involve the removal of stabilizing cross-links or of the necessary amino acid residues, leading to a loss of bioactivity. Therefore, an interest developed in structural modifications in the form of mixed peptide hybrids, in which conformationally constrained yet irrelevant amino acid residues are replaced with isosteric spacers. An advantage of this minimization strategy is that the resulting molecules retain the overall size of the original molecule and thus do not disturb the spatial orientation of key functional residues. Replacing parts of the peptide backbone with different non-peptide spacers improves bioavailability by reducing the susceptibility to proteolysis and the number of hydrogen bond donors/acceptors. One such isosteric fragment is 6-aminohexanoic acid, which provides chain flexibility to its methylene bridges without losing the parent’s biological activity [32].

Terminally blocked tripeptides containing 6-aminohexanoic acid, and Gly, Aib, Leu and Phe residues form supramolecular β-sheets in the solid state, similarly to how amyloid-like fibrils, such as Ahx, use residues in which the two-end moieties, i.e., NH and C=O, can provide hydrogen-bonding functionalities, while the centrally located pentamethylene unit provides sufficient flexibility to the peptide backbone to mediate the supramolecular sheet aggregation and the subsequent amyloid-like fibril formation of an extended backbone conformation [33]. The molecular basis of fatal neurodegenerative diseases involves an intermolecular sheet aggregation and the subsequent formation of highly ordered fibrillar structures.

Research into new DNA cleaving agents is of great importance in molecular biology. It has been shown that simple Lys-X-Lys tripeptides (where X is an aromatic residue) can bind to polydeoxyribonucleotides, and native and denatured UV-irradiated DNA through aromatic rings interposed between bases, and through ionic bonds, i.e., positively charged lysyl side chains with phosphates. In addition, a specific single-strand cleavage of the supercoiled plasmid has been reported for the Lys-Trp-Lys and Lys-Tyr-Lys tripeptides. Cyclic peptides of this type, using Ahx as a linker, confirmed the stereochemical matching of the small-sized ring of the cyclic peptides to the smaller DNA groove, compared to the corresponding linear peptides [34]. Cyclic peptides do not have a C-terminal carboxylate function. They are expected to have a higher DNA-binding affinity, as the anionic carboxylate group present in the linear peptides exhibits electrostatic repulsion with the anionic phosphate backbone of DNA, and the methylene groups in the Ahx residue provide additional hydrophobic interaction with DNA. The results show that the Trp cyclic tetrapeptide is a potential DNA cleavage agent with a higher apparent DNA binding constant as well as a higher cleavage rate than the parent linear peptide, at temperatures below 60°C.

The effect of stereochemistry on the ability to cross biological barriers has also been examined. The L,L- and L,D-Ala-Ala dipeptides were cyclized with Ahx to give macrocycles in which the dipeptides correspond to positions i + 1 and i + 2 of the beta turn [35]. The transport of the dipeptides through the Caco-2 cell monolayer was determined, along with the corresponding acyclic models. Measurements performed in the presence of verapamil showed that the cyclic peptides experienced approx. 4-5-fold difference in internal flux, depending on the stereochemistry, with the L,D isomer being transported at a higher rate.

Simple dipeptides cyclized with 6-aminohexanoic acid or its derivatives have been investigated as small-molecule models of β-turns [35]. Due to the well-known difficulties in using peptides as oral therapeutic agents, peptides containing modified amino acid residues or non-peptide fragments are synthesized. 6-aminohexanoic acid is a bioisoster introducing an element into the structure that facilitates the assumption of the desired conformation. The β-turn, a nonrepeating motif, is the third most common secondary structure in proteins, behind the α-helix and β-sheet. The definition of the most common β-turns requires that the amide bonds reside in the trans orientation and that the distance between the a-carbon of the i and i + 3 residues be <7 Å. Ahx is the best candidate to act as the “third” amino acid in a cyclic peptide.

Deamidation of asparagine (Asn) residues in proteins, which introduces a mutant aspartate (Asp) or an iso-aspartate (isoAsp) residue at the Asn position, is an in vivo pathological aging-related event. Moreover, due to the occurrence of this phenomenon during the storage of biotechnological pharmaceuticals, it constitutes a barrier to the use of proteins and peptides as therapeutic agents. The flexible structure of Ahx facilitates the arrangement of the peptide structure in the β-sheet. N-acetyl-Asn-Gly-6-aminocaproic acid, an acyclic β-turn mimetic analog, deamidates the aspartyl residue to produce N-acetyl-Asp-N-Gly-6-aminocaproic acid about three times faster than the cyclic β-turn mimics the cyclo[L-Asn-Gly-Aca-] with asparagine at position 2 of the β-turn [36]. The latter, in turn, deamidates ~30 times faster than its cyclo[Gly-Asn-Aca-] positional isomer with asparagine at the 3-position of the β-turn. Both cyclic peptides adopt mainly β-turn structures in solution, as demonstrated by NMR and circular dichroism characteristics. The open-chain compound and its N-acetyl-Gly-Asn-Ahx isomer assume predominantly random coil structures and therefore deamidate twice as slowly as the former [34].

Although aminohexanoic acid is commonly used in peptides, the first example of a successful introduction of Ahx into a conformationally constrained peptide was Ahx-SIIIA, an analog of conotoxin SIIIA, with Ahx used as an isosteric replacement of nonessential amino acid residues in the conformationally constrained part of a three disulfide-bridged peptide [37]. Compared to SIIIA, the new peptide proved to be a better inhibitor of sodium currents in the dorsal root ganglia and sciatic nerves in mice and a more potent and more durable analgesic in an inflammatory pain model in mice.

Prostheses of non-critical parts of a polypeptide were applied to an opioid neuropeptide, Met-enkephalin, in which two adjacent Gly2-Gly3 residues were replaced by ε-Ahx spacers [38]. Backbone spacers did not affect the general structural properties of the analogs, but radically reduced their affinity and agonist activity towards δ- and μ-opioid receptors. Molecular modeling suggested that the decrease in Met-enkephalin affinity for the opioid receptor could be explained by the loss of a single hydrogen bond. Interestingly, the analogs containing the most isosteroid spacers retained strong antinociceptive and anticonvulsant properties, comparable to those of the endogenous peptide. This unexpectedly high in vivo potency could not be explained by the increase in metabolic stability. The results obtained with analogs containing backbone linkers suggest a new mechanism of seizure control in the brain, one that involves alternative non-opioid signaling.

This strategy has also been applied to other peptides, such as neuropeptide Y (NPY) [39]. The affinity for the Y1 receptor was investigated on the human neuroblastoma cell SK-N-MC, and the prolongation of the Ahx to N-terminus peptide led to the analog that exhibited the best Y1 receptor affinity.

6-aminohexanoic acid can also play the role of a hydrophobic linker between charged residues in peptide molecules. Steric-block oligonucleotides, i.e., peptide nucleic acids or phosphorodiamidate morpholino oligomers, are uncharged DNA-mimics, poorly taken up by most cell types; conventional delivery strategies that rely on electrostatic interaction do not apply. The insertion of one ε-Ahx molecule between two arginine (R-Ahx-R)_4_ molecules causes separation of the guanidine groups in the cell-penetration peptide (CPP) oligoarginine and is a key factor for cellular internalization and metabolic stability. This leading CPP is being developed for therapeutic use to enhance penetration of synthetic oligonucleotides through the cell membrane in the production of a functional fully spliced mRNA in the treatment of diseases, such as β-thalassemia, Duchenne muscular dystrophy, or cancers [40,41,42,43]. For example (R-Ahx-R)_4_-PNA705 (PNA = NH_2_-Cys(NPys)-Lys-CCTCTTACCTCAGTTACA-Lys-amide) conjugate with Ahx are capable of redirecting splicing more efficiently than conjugates of Tat, Penetratin or oligo-Arg classical CPPs in HeLa pLuc705 cells [44].

The coupling of 6-aminohexanoic acid with short synthetic peptides (5-10 amino acid residues) resulted in enhanced hydrophobicity of the compounds and improved coating efficiency in the ELISA immunoassay procedures [45]. To improve the attachment of the peptide to the solid phase of the polystyrene plates, the use of (Ahx)_2_T^127^FIQFKKDLKEW^137^(Ahx)_2_ was examined in place of poly-L-lysine or poly-L-aspartate. By using a microplate coated with the modified peptide, a several-week-long stability at 4 °C was reached, reducing the time and cost of the test.

The presence of ε-Ahx residues in the structure of a peptide can prevent enzyme hydrolysis in vivo and influence its biological activity. The insertion of 6-aminohexanoic acid between histidine and alanine at positions 7 and 8 of glucagon-like peptide 1 (GLP-1) prevents N-terminal degradation by dipeptidyl peptidase IV and produces an effective, long-acting analog of the polypeptide, which may be useful in the treatment of type 2 diabetes mellitus. The insertion of 4 and 8 ε-Ahx moieties between positions 8 and 9 of GLP-1 results in a reduction of biological activity, due to the reduced affinity for the receptor [46].

Ahx has also been inserted into the structures of potential low molecular enzyme inhibitors with a peptide structure designed on the basis of a fragment of human angiotensinogen, e.g., renin inhibitors [47]. The compound, containing Ahx-Iaa at the P2’–P3’ Boc-Phe(4-OMe)-MeLeu-AHPPA-Ahx-Iaa (IC_50_ = 1.05 × 10^−9^ M), could enter into hydrophobic interactions at the S2′–S3′ site of the enzyme.

## 5. Ahx as a Linker

Linkers, also called spacers, are flexible molecules or stretches of molecules that are used to link together two molecules or fragments of molecules. Naturally occurring separators, e.g., in proteins, are Gly-rich linkers [48], which allow distinct domain functions. Separators are very flexible and retain or enhance the biological activity of modified molecules. The linker does not affect any constraints on the conformation or interaction of the linked fragments.

6-aminohexanoic acid is used as a linker (Figure 3) in the peptide chain of new cyclic peptides containing the 9–22 epitope {c[CH_2_CO-^9^LKMADPNRFRGKDL^22^AhxK(Ac-CGFLG)AhxC]-NH_2_}_2_ sequence derived from the herpes simplex virus (HSV) type 1 glycoprotein D (gD-1) [49]. The binding of a monoclonal antibody, Mab A16, to the synthesized compounds was determined by enzyme-linked immunosorbent assay. It was demonstrated that cyclization decreased the binding activity of the antibody to the epitope. However, dimerization and conjugation could significantly increase the binding capacity of the cyclic epitope peptides. The attachment site in dimers and conjugates, as well as the topology of the construct, had a significant influence on the antibody recognition.

c[CLNSMGQDC-6-aminohexanoic acid-CLNSMGQDC] is a cyclic peptide in the linking of N- and C-terminal, which are two cysteine residues in the generation of tandem peptides, synthesized as a new therapeutic approach for the treatment of vascular hyperpermeability through the stabilization of vascular endothelial (VE) cadherin-mediated adhesion [50].

ε-Ahx has also been used to link arginine molecules to the poly(amidoamine) dendrimer (PAMAM) structure to modify its hydrophobicity and flexibility [43]. PAMAM is widely tested as a non-viral vector, both in vitro and in vivo, as a gene and drug delivery system. 6-aminohexanoic acid was introduced into the dendrimer as a hydrophobic spacer between the PAMAM G4 dendrimer primary amine groups and the terminal L-Arg. PAMAM G4-Ahx significantly reduced the efficiency of transfection, and the additional coupling of Arg residues to the dendritic N-terminus of Ahx chains (PAMAM G4-Ahx-Arg) resulted in an enhanced effect. In addition, compared to the derivative containing only arginine, the synthesized derivative containing Ahx was significantly less cytotoxic to the 293 tested human embryonic kidney cells.

Acridine derivatives are characterized as a new class of compounds with an increased ability of insertion between DNA base pairs with potential anti prion activity. A suitable spacer was responsible for stability, contact and binding to domains. The group of compounds, created by Borgions, contains acridine in the central molecule and long peptide chains [51,52]. The length and flexibility of the peptide assures binding to three DNA domains. A good linker is the 6-aminohexanoic acid. Ac-Arg-Tal-Chi-Chi-Cbg-Cha-Arg-6-Ahx-Lys(Acr)-6-Ahx-Arg-Cbg-Cha-Chi-Chi-Tal-Arg-NH_2_, has a binding affinity of 0.74 × 10^−5^ M, compared to the known minor-groove binder Hoechst 33258, with its 5′-AAAAA-3′ target sequence with a Kd value of 1.0 × 10^−6^ M.

ε-Ahx has been used in the chimeric strategy to design new analogs of peptide hormones by combining fragments of arginine vasopressin (AVP) and bradykinin (BK) in order to obtain AVP(1-9)-ε-Ahx-BK(1-9) [53] and other chimeric peptides as potential biological agents with high affinities for the bovine kidney medulla B2, bradykinin receptor.

One option for improving the biological properties of the compounds is to use an Ahx in place of a peptide fragment to design synthetic peptide/protein mimics to change in the α-helical conformation and the orientation of the side chains of amino acids. According to this approach, one of the most novel neuropeptide Y (NPY) analogs is NPY-(1-4)-Ahx-(25-36), in which the N-terminal tetrapeptide is linked to the C-terminal dodecapeptide amide via the 6-aminohexanoic acid, thus ignoring the rest of the NPY 5-24 loop. In several tests, this analog shows an affinity for the NPY receptor on rabbit kidney membranes comparable to that of the native hormone, as well as biological activity, including an increase in blood pressure in intact rats [54].

Two seven-residue helical segments, Val-Ala-Leu-Aib-Val-Ala-Leu, have been linked synthetically with an ε-aminocaproic acid linker with the intention to make a stable antiparallel helix–helix motif [55]. The joining of two helical segments by a flexible linker residue Ahx, should provide hydrogen bonding groups with N-terminal NH and C-terminal C=O that serve to continue the helical conformation of the individual modules, while the central (CH_2_)_5_ hydrocarbon chain should provide sufficient flexibility for chain folding. De Rosa et al. reported the design of a set of peptides mimicking two secondary structure elements of VEGF (α-helix 17-25 and β-hairpin 79-93) involved in receptor recognition [56]. The two linear amino acid segments were synthesized by the chemical ligation reaction through variable amino acid spacers to check the impact of length and flexibility on the biological activity. Of those investigated by NMR techniques, the ε-aminocaproic acid conjugate, among others, was the most biologically effective peptide interacting with the recombinant second domain of VEGFR1.

The same strategy was used for rational structure minimization of fragment 1-45 of eosinophil cationic protein (ECP), retaining the antimicrobial activity of the whole protein. Structure-based sizing focused on both α-ECP helices (1–45) and gave analogs with significant potencies against gram-negative and -positive strains. Compared to (1–45), ECP analogs (8–36) and ECP (6–17)-Ahx-(23–36) provide a size reduction of 36% and 40%, respectively, and exhibit an antimicrobial profile remarkably similar to that of ECP. Both retain the segments required for self-aggregation and lipolisaccharide binding as well as the ability to agglutinate the parent ECP bacteria. In particular, the analog of (6–17)-Ahx-(23–36) is shown by NMR to preserve the spiral features of the native 8–16 (α1) and 33–36 (α2) regions and can be proposed as the minimum structure capable of restoring the activity of the whole protein [57]. The mechanism of action of ECP on the surface of bacteria (agglutination) is very considerably influenced by the structure of LPS, in particular its polysaccharide moiety, as confirmed by the described peptide from Ahx [58].

An antisense oligodeoxynucleotide-doxorubicin conjugate has been synthesized by an aminocaproic acid linker. The conjugate showed a remarkable stability as compared to the unmodified oligodeoxynucleotide. The result showed that the conjugate displayed a very high reversal multidrug resistance activity in KB-A-1 cells in vitro. The conjugate lowered the IC_50_ value from 21.5 mM to 2.2 mM, with a fold-reversal factor of 10. In contrast, a slight decrease of the IC_50_ value was observed when it was combined with “free” antisense oligodeoxynucleotide: the IC_50_ value was down from 21.5 mM to 16.8 mM. This study suggests that antisense oligodeoxynucleotide-doxorubicin conjugate with Ahx might be helpful in multidrug resistance reversal [59].

A large group of compounds containing Ahx as a linker are compounds used in cancer phototherapy and as auxiliary compounds in the construction of probes in microscopic localization of single molecules. The presence of Ahx in the structures of the compounds given below does not result in an improvement of biological activity. However, it influences the improvement of some properties of potential therapeutics, such as solubility and permeability through cell membranes.

The selectivity of photodynamic therapy (PDT) as an anti-cancer treatment is based on the local production of cytotoxic reactive oxygen species in the tumor as a result of both the preferential uptake of the photosensitizer (PS) by the tumor tissue and the subsequent local light irradiation. However, there are some limitations and inherent disadvantages of using PS (first generation), such as poor chemical purity, short wavelength of light, extended half-life, and intense accumulation in normal tissues due to photosensitivity (Figure 4) [60]. One way to solve this problem is to conjugate or encapsulate the existing PS in carriers (conjugation of PS with new synthetic peptides) that can deliver therapeutic agents to the target tissue with lowered unspecific targeting of healthy tissues.

ε-Ahx has been used to combine a photosensitizer, 5-(4-carboxyphenyl)-10,15,20-triphenylchlorine (TPC), with ATWLPPR heptapeptide (TPC-Ahx-ATWLPPR), which has an affinity for endothelial tumor cells by targeting the vascular endothelial growth factor receptor (VEGF165) neuropilin-1 (NRP-1) rather than the VEGF type 2 receptor (VEGFR2/KDR) [61] and conjugates to TKPRR or DKPPR designed to act on the neuropilin-1 receptor (NRP-1) [60]. A competitive binding study comparing unconjugated and conjugated TPC showed that conjugated TPC (conjugated to TKPRR, DKPPR or ATWLPPR) successfully binds to the NRP-1 receptor and has the ability to shift VEGF165 binding, while unconjugated TPC does not.

Additionally, the obtained conjugates increase tissue accumulation and are much stronger photosensitizers (values of 50 light doses after incubation with TPC-Ahx-ATWLPPR or TPC were 0.47 ± 0.23 and 4.9 ± 0.64 J/cm^2^, respectively). The binding of biotinylated VEGF165 to NRP-1 was displaced by new conjugates (TKPRR or DKPPR) in a concentration-dependent manner. This showed that the spacer length and nature did not affect the affinity of the peptides for its receptor; instead, the spacers were found to increase the solubility of the conjugates. TPC–Ahx–ATWLPPR was incorporated up to 25-fold more in human umbilical vein endothelial cells (HUVEC) than TPC. The new conjugates also showed better affinity for NRP-1 compared to TPC-Ahx-ATWLPPR (IC_50_ = 171 µM) [60,61]. The use of the Ahx linker provides a space between the TPC and the peptide so that they are individualized and separated; it also gives the molecule flexibility [62].

Research into synthesized analogs of the thermostable human *Escherichia coli* peptide (STh) containing the DOTA chelating group is also promising. The binding affinity of the synthesized STh analog for GC-C receptors in human T-84 colon cancer cells showed that DOTA-6-Ahx-Phe19-STh binds to the GC-C receptors present in these cells with high specificity and affinity, and higher cytotoxicity than analogs without the Ahx group (IC_50_ 1.6 nM) [63]. The results of these studies show that radiometallated DOTA-STh analogs are promising candidates for further development as diagnostic and/or therapeutic radiopharmaceuticals for use in colorectal cancer patients.

In the case of prostate cancer, identification biomarkers, i.e., gastrin-releasing peptide receptors (GRPr) and prostate-specific membrane antigen (PSMA), are extremely important. The receptor-binding affinity was assessed in human prostate cells, PC-3 (GRPr-positive) and LNCaP (PSMA positive), and the tumor targeting efficacy was determined in severe combined immunodeficiency (SCID) mice using DUPA-6-Ahx-(^64^Cu-NODAGA)-5-Ava-BBN(7-14)NH_2_. MicroPET molecular images clearly show the effectiveness of the application of the multivalent GRPr-/PSMA targeting the radioligand [64,65]. Recently, there has been growing debate as to whether agonist or antagonist ligands are better radiopharmaceuticals to target GRPR. While biodistribution studies explain the higher tumor uptake and the better tumor/non-tumor ratio for the antagonist ligand, microPET/CT images clearly confirm that the agonist is a much better GRPR-targeted molecular imaging agent. High-quality, high-contrast images obtained with the ^64^Cu-NODAGA-6-Ahx-BBN(7-14)NH_2_ agonist are comparable to those of the vector targeting the ^64^Cu-CB-TE2A-AR antagonist, which is currently undergoing clinical trials in Europe [66].

Ahx acid has also been used as a linker for the construction of probes in SMLM (microscopic localization of single molecules)—MGVADLIKKFESISKEEGGGGK(Ahx-FITC)GGrRrRrRRR. Single-molecule localization microscopy (SMLM) provides super-resolution imaging beyond the diffraction limit, but is critically based on the use of photomodulated fluorescent probes that have the ability to deliver the cytosolic cell-penetrating peptide (rR)_3_R_2_. Fluorescent probes consist of a (rR)_3_R_2_ peptide linked to a cell-impermeable organic fluorophore and recognition unit. The probe, MGVADLIKKFESISKEEGGGGK(Ahx-FITC)GGrRrRrRRR, shows excellent cell permeability and high specificity; however, after incubation with live BSC-1 cells, no photo-blink was observed [67]. Therefore, under such circumstances, it is not suitable for super-resolution imaging of live cells.

## 6. Ahx as an Element of Biotinylation Reagents

Biotinylation is one of the most widely used protein labeling methods to facilitate detection, immobilization and purification. Biotin is a naturally occurring compound in organisms containing a system of fused imidazolidine and thiolate rings with a 4-atom alkyl chain terminated with a carboxyl group. It is a carboxylase coenzyme that participates in the transfer of the carboxyl group to various organic compounds. Biotinylation is employed in many areas of biotechnology, specifically those which use fast and specific binding to avidin and streptavidin, and does not interfere with the natural functions of molecules due to the small size of biotin (MW = 244.31 g/mol) [68,69,70]. Most chemical biotinylation reagents consist of a reactive group attached via a linker to the biotin side chain of valeric acid. Biotinylation reagents having a longer linker (>4 Å, ~5 atoms) are desirable because they allow the biotin to reach the avidin/streptavidin binding pocket located approximately 9 Å below the protein surface [71]. Increasing the length of the carbon chain also reduces the steric hindrance of the biotin rings and facilitates access of the reactive linker, enabling further reactions used in the analysis or in the synthesis. The currently commercially available Ahx derivative in combination with biotin is Fmoc-Lys(biotinyl-ε-aminocaproyl)-OH, used in solid phase peptide synthesis [72,73,74,75,76,77].

Ahx with biotin residues has been used in the structure of synthetic substrates of tumor-associated endopeptidases from colorectal tumor. Biotin-Ahx-ERGFFYPHHHHHH and Biotin-Abu-Ahx-WKPYDAADL-Ahx-HHHHHHt increased the diagnostic sensitivity for mass spectrometry-based protease profiling of serum specimens for improved disease classification [78].

Three types of biotinylated peptides with different linkers between biotin and the β-sheet peptide are designed and synthesized [79]. The PKFKIIEFEP peptide exists as individual single fibers and does not form a gel at peptide concentrations of less than several millimolar in aqueous solutions near neutral pH. Each of the biotinylated peptides, however, forms a tubular structure. The most hydrophobic linker is composed of two units of 6-aminohexanoic acid, and the most hydrophilic linker consists of two units of 8-amino-3,6-dioxaoctanoic acid. The middle linker was 8-amino-3,6-dioxaoctanoic acid linked with 6-aminohexanoic acid. Transmission electron microscopy shows that biotinylated peptides self-assemble to form a nanotube construction with an outer diameter of approximately 60 nm and an internal diameter of approximately 30 nm in aqueous solution. Antibiotin, i.e., the antibody, effectively binds to biotin groups on the peptide nanotubes. Antibody binding is regulated not only by protein concentration in the solution, but also by the properties of biotinylated tube peptides. Antibody preferentially bound to biotinylated peptide tubes, which consist of a peptide with the most hydrophilic linker, suggesting that the properties and functions of the surface of the tubular structure are modulated and designed by the peptides.

Ahx linker has been used in hepcidin derivatives [80]. N-terminal (+)-biotinyl-6-aminocaproic acid-DTHFPIC and CCHRSKCGMCCKT derivatives of 25-mer hepcidin were adequate representatives of the molecule in immune-adsorption-based assays (ELISA, immunodot analysis) using a commercially available polyclonal antibody. The conjugated biotin moiety binds one molecule of streptavidin, resulting in signal amplification. The binding of biotin and streptavidin is one of the strongest noncovalent interactions found in nature. Because the binding sites for biotin are buried deep inside the streptavidin structure (about 9 Å below the protein surface), spacers improve both the accessibility and the reaction rate of biotinylated compounds with respect to streptavidin, often enhancing the sensitivity of assay systems. The most common spacer group is 6-aminocaproic acid, which increases the distance of the side arm by about 9 Å. In the case of N-(+)-biotinyl-6-aminocaproic acid compound, there is a C6-spacer between the biotin and the peptide molecule; hence, it reduces the steric hindrance and increases the accessibility of streptavidin. Peptidyl diazomethane cysteine inhibitors labeled with biotin proteinases, based on an N-terminal segment similar to the substrate of human cystatin C—a natural inhibitor of cysteine proteinases—have been synthesized. These synthetic derivatives were tested as irreversible inhibitors of cruzipain, the main cysteine proteinase of Trypanosoma cruzi, to compare the inhibitory kinetics of the parasitic proteinase with that of the mammalian cathepsin B and L. The inhibition of cruzipain by Biot-LVG-CHN3 and Biot-Ahx-LVG-CHN_3_ was similar to an unlabeled inhibitor. The biotin labeling of the inhibitor slowed down the inhibition of both cathepsin B and cathepsin L. The addition of a spacer arm between the biotin and the peptide derivative moiety increased the inhibition of cathepsin B but not cathepsin L. The discrimination provided by the spacer is likely due to the differences in the topology of the proteinase binding site, a feature that can be used to improve the targeting of individual cysteine proteinases [81].

The synthesis of a new cyclic hexapeptide specific for RIIb-3 integrin that contains the Arg-Gly-Asp (RGD) sequence and is coupled to a dimyristoylthioglyceryl anchor has been described [82]. With the biotinylated peptide analog, the optimal-length spacer was searched for between the peptide and lipid moieties by assessing the binding strength using an enzyme-linked immunosorbent test (ELISA) and by surface plasmon resonance (SPR). It was found to be strongly dependent on the length of the spacer introduced between the biotin and the peptide residues of ligands that consisted of Ahx or Ahx with two additional units of glycine. The best results were obtained with c[Arg-GlyAsp-D-Phe-Lys(Biot-Ahx-Gly-Gly)-Gly-] with KD (dissociation constant) 0.158 µM from ELISA and KD 1.1 µM from SPR measurements. A new integrin-specific ligand makes it possible to establish new model systems for systematic studies of integrin cluster self-organization and focal adhesion complexes.

A biotin derivative, namely biotin-aminocaproic acid-lysine (BAL), was conjugated to a carrier-protein and used for rabbit immunization [71]. The aminocaproic acid-lysine “long-arm” was used in order to project the biotin-hapten above the carrier-protein surface. Lysine was selected due to its N^ε^-amino group, through which BAL was conjugated to the carrier-protein. The anti-BAL antibodies recognized free biotin, as shown with an in-house-developed ELISA, in which biotin conjugated to a synthetic “lysine-dendrimer” was used to coat the ELISA microwells. In immunocytology and Western blot experiments, the anti-BAL antibodies led to results similar to those obtained with streptavidin. Synthetic derivatives of hapten molecules that can be easily prepared with solid-phase chemistry, such as BAL, may be used for the development of specific antibodies for the corresponding hapten.

The coupling of ferrocene moieties to avidin via a flexible spacer molecule, Ahx, yields a conjugate, which combines the unique biotin-binding properties of avidin with the reversible redox characteristics of ferrocenes [83]. The spacer chain bears a carboxylic acid functionality to be used for coupling to proteins. Covalent immobilization of the conjugate on gold electrodes in a dense monolayer results in electrodes with a high binding capacity for biotinylated molecules as well as good electron transfer properties. The application potential of such electrodes for bioelectrochemical systems is demonstrated by electrochemical reduction of hydrogen peroxide under mild conditions catalyzed by a bound biotin-microperoxidase MP11 conjugate.

A linker composed of two 6-aminohexanoic acid fragments has been used to synthesize the modified dye-labeled nucleotide and to examine the effect of this modification on the incorporation by thermophilic DNA polymerases [84]. Cy5-dUTP and a variant nucleotide in which the linker had been lengthened by 14 atoms between the dye and the nucleobase were compared. It was found that the Cy5-dUTP with a longer linker resulted in longer primer extension lengths.

A galiellalactone biotinylated analog has been used in various biological studies to elucidate the mechanism of action of galiellalactone, starting with its effect on STAT3 signaling. It was found that they inhibit the proliferation of DU145 cells expressing constitutively active STAT3 and demonstrate that the attachment of biotin via a Ahx linker produces cell-permeable covalent STAT3 inhibitors that can be used as important tools for elucidating the mechanism of action of galiellalactone [85].

## 7. Polymers and Oligomers of Ahx

6-aminohexanoic acid is the only monomer of the nylon 6 polyamide polymer where the ω-amino acid molecules are connected by amide bonds. It is also one of the monomers of other synthetic polymers with a name that uses the acronym PAHBAH [86], PAHBAT [87], where AH, B, A, T are the representative characters of the different units: aminohexanoic acid, 1,4-butanediol, adipic acid, and terephthalic acid, respectively.

6-aminohexanoic acid can easily form cyclic lactam derivatives. The monomer, namely 6-hexanelactam (ε-caprolactam), is a substrate in nylon-6 synthesis. Cyclic oligomers are the by-products of the process. The tendency of these cyclic structures to be hydrolyzed decreases in the following order: monomer >> trimer > tetramer >> dimer. The cyclic dimer (1,8-diazacyclotetradecane-2,9-dione) has a very poor solubility in common solvents and low reactivity. Its very slow hydrolysis has been explained by the high conformation stability of its fourteen-membered cycle and by the mutual effect of two amide groups [88]. These properties are often considered to be responsible for the difficult processing of the oligomer mixture sourced from industrial extracts remaining as waste from the production of polyamide nylon-6. However, bacteria that can grow on a medium containing 6-aminohexanoic acid cyclic dimer as the sole source of carbon and nitrogen, was isolated from waste-water sourced from a nylon factory. These strains of bacteria can produce enzymes [89,90] that can hydrolyze cyclic oligomer bonds. 6-hexanelactam has also been used as a substrate in the catalytic synthesis of low molecular weight oligomers of N-acyl-6-aminohexanoic acid [91,92]. Copolyamino acids based on 6-aminohexanoic acid and protein amino acid structures have been investigated as potential biodegradable polymers [93,94] that can be used for biomedical purposes and as biodegradable packing materials. For example, copolymers of 6-aminohexanoic acid and hydroxyproline have been investigated as potential bone repair materials [95].

## 8. Other Derivatives

Other (i.e., not mentioned above) derivatives of ε-Ahx that exhibit biological activity also deserve attention. This group includes esters and amides of 6-aminohexanoic acid and its substituted derivatives that increase transdermal penetration and thus facilitate drug delivery [96]. In vitro studies of the skin permeation enhancing effect of long-chain alkyl and N-alkylamide esters of 6-aminohexanoic acids with an acyclic or cyclic tertiary amine group and alkyl chains ranging from octyl to dodecyl have been carried out on cut human skin strips with theophylline as a model permeation agent. All the tested substances showed a strengthening effect, while six (pyrrolidin-1-yl)decylhexanoate showed the highest activity (ER = 30). The results obtained in the study show that the esters were stronger than the amides, and the increase in the volume of the final basic substituent led to a decrease in activity (the optimum basicity existing in the pKa region was slightly below 9) [97,98,99]. In addition, derivatives of tranexamic acid (an ε-Ahx analog) were investigated. Ethyl 6-(2,5-dioxopyrrolidin-1-yl)-2-(2-oxopyrrolidin-1-yl) hexanoate (3) showed the lowest lipophilicity (3), while the highest was observed for dodecyl-6-(2,5-dioxopyrrolidin-1-yl)-2-(2-oxopyrrolidin-1-yl) hexanoate tranexamic acid (TXA) derivatives, especially those with 10 and 12 carbon chains. Increased skin permeability for both simple hydrophilic and lipophilic was confirmed by in vitro studies. The idea of possible enhancement of the capabilities of TXA derivatives was based on the similarity between TXA and 6-aminohexanoic acid, which is a component of transcarbam 12, a highly effective skin permeation enhancer [99].

Moreover, 6-aminohexane hydroxamate (HC6) induced differentiation in mouse neuroblastoma cells at submillimolar concentrations. Interestingly, other hydroxamates with different chain lengths were not as effective as HC6. HC6 induces neurite outgrowth and expression of neuron-specific genes, such as synapsin I and MAP-2 in neuroblastoma cells, in the absence of other promoting factors, such as cAMP. The effect of HC6 on neuroblastoma cell differentiation was comparable to, or even better, than that of N6, O2′-dibutyryl cAMP (Bt2cAMP), a standard reagent commonly used to induce the differentiation of murine and human neuroblastoma cells in culture [100]. Danazol-aminocaproic derivatives are also of great importance, as they exhibit antibacterial properties. Experimental data suggests that the molecular mechanism involved in the antimicrobial activity induced by the danazol derivative may be due to the fact that the steroid requires a hydrophilic spacer arm region with a free carboxyl group to interact with some cell surface factors, integrate into the cytoplasmic membrane, and consequently lead to the induction of the inhibition of bacterial growth [101].

## 9. Conclusions

6-aminohexanoic acid is a well-known drug still used in therapy due to the lack of an effective alternative. It is cheap and easy to synthesize. It is a monomer used for the production of polyamide polymers. Its biological activity and medical use so far has only concerned antifibrinolytic properties. However, due to its particularly flexible structure, its hydrophobic nature, and the distance between the carboxyl and amino terminal groups, it can potentially be used to modify the structure of other molecules, or as a spacer linking fragments of active molecules. As the examples of Ahx-modified molecules cited in the above paper show, this fragment is not necessarily responsible for biological activity, but it still significantly improves the interaction with the molecular target.

## Figures and Tables

**Figure 1 ijms-22-12122-f001:**
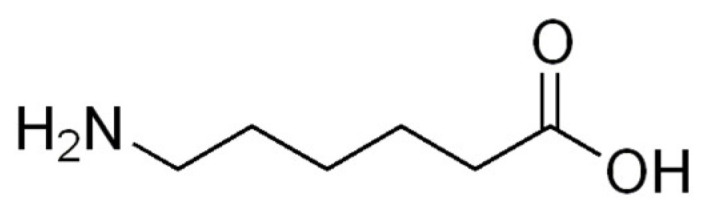
Structure of 6-aminohexanoic acid.

**Figure 2 ijms-22-12122-f002:**
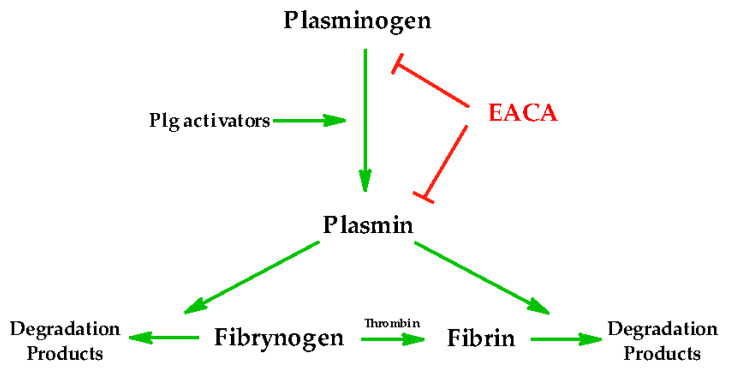
Scheme of the plasminogen–plasmin system. Inactive plasminogen is converted into plasmin as a result of an action of plasminogen activators (natural: urokinase plasminogen activator or tissue plasminogen activator, or synthetic: streptokinase). Plasmin then enables the degradation of fibrinogen (a plasma protein) as well as the degradation of the fibrin clot into fibrin cleavage products. The unbound circulating fibrinogen is converted into fibrin clot by thrombin. The formation of plasmin from plasminogen can be inhibited by antifibrinolytic drugs (Ahx). The interaction of Ahx with plasminogen or plasmin, as well as plasmin with fibrin or fibrinogen, is mediated by lysine binding sites, not enzymatic active sites of plasmin. Green arrows = activation, red arrows = inhibition.

**Figure 3 ijms-22-12122-f003:**
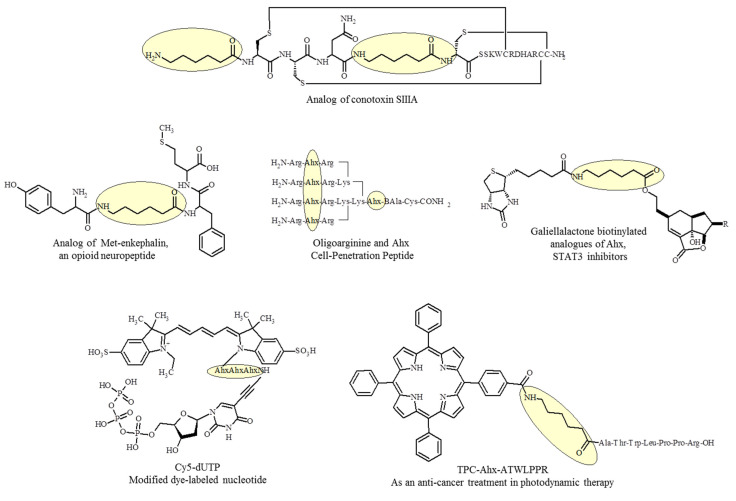
Ahx as fragment of molecules with biological activity.

**Figure 4 ijms-22-12122-f004:**
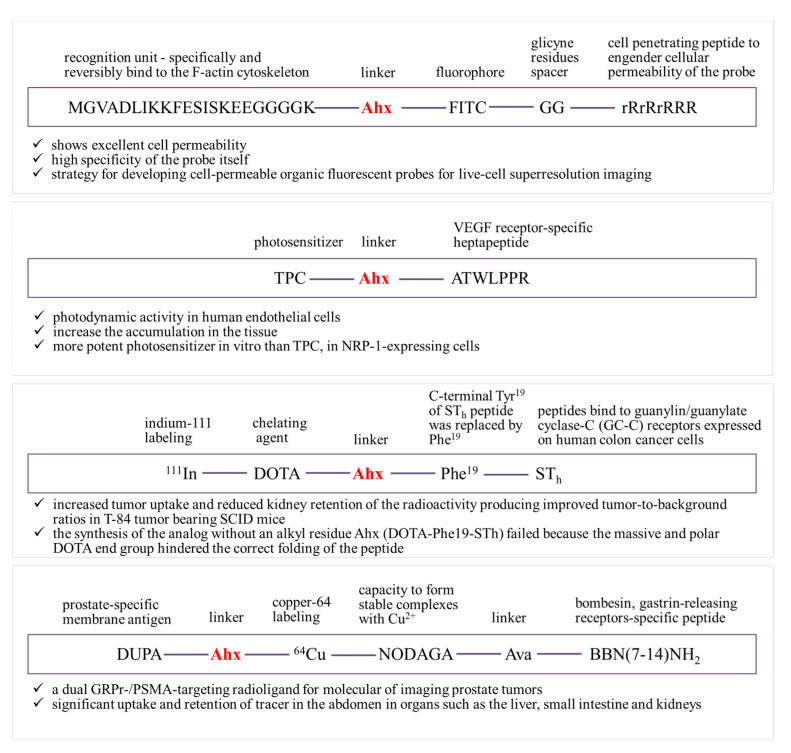
Examples of structures using Ahx as linker in receptor affinity peptide structures.

**Table 1 ijms-22-12122-t001:** Biological activity of peptides analogs of Ahx.

Structure	Antifibrynolytic Activity IC_50_ [mM]	Antiamidolytic Activity/S-2251 IC_50_ [mM]	References
EACA	0.2	-	[20]
H-EACA-NLeu-OH	<0.02	0.12	[21,22,23,24,25]
HCl × H-EACA-Leu-OH	0.08	13	
HCl × H-EACA-Leu-OEt	-	0.2	
HCl × H-EACA-Cys(S-Bzl)-OH	0.04	18	
HCl × H-EACA-Cys(S-Bzl)-OEt	-	12	
H-EACA-EACA-OMe	-	0.16	[26,27]
HCl × H-EACA-Gly-OH	0.8	8	
HCl × H-EACA-Glu-OH	-	0.08	
Boc-EACA-Lys(Z)-EACA-NH_2_	20	20	[28]
Boc-EACA-Lys-EACA-NH_2_	19	0.02	
H-EACA-Lys-EACA-NH_2_	8.1	9	
Boc-EACA-Lys(Z)-EACA-OMe	1.2	>20	
H-EACA-Lys-EACA-OMe	18	0.8	
Boc-Lys(Z)-EACA-NH_2_	<0.2	8	
Boc-Lys-EACA-NH_2_	<0.2	-	
H-D-Ala-Phe-Lys-EACA-NH_2_	-	0.02	[29]

**Table 2 ijms-22-12122-t002:** Ahx introduction into the structure of peptides with biological activity (the original abbreviations of 6-aminohexanoic acid used in the references were used in the table).

Structure with Ahx	Biological Activity Benefits of Inserting of Ahx	References
Boc-Acp-Aib-Phe-OMe	Peptides form hydrogen bonded dimers that gives supramolecular β-sheets on self-assembly.	[34]
cyklo[Lys-Tyr-Lys-Ahx-] cyklo[Lys-Trp-Lys-Ahx-]	Higher DNA binding constant than linear analog.	[35]
cyclo(L-Ala-L-Ala-Aca) cyclo(L-Ala-D-Ala-Aca)	Cyclization facilitates penetration through the Caco-2 human epithelial cancer cell line monolayer.	[36]
cyclo-[L-Asn-Gly-Aca] cyclo-[Gly-Asn-Aca]	Cyclic peptides assume predominantly β-turn structures in solution, faster deamidation than linear analogs.	[37]
Ahx-SIIIA	Improve bioavailability by reducing susceptibility to proteolysis, and reducing hydrogen bond donors/acceptors.	[38]
Y-(6-Ahx)-Phe-Met	Reduction of affinity and agonist activity towards δ- and μ-opioid receptors, while maintaining strong antinociceptive and anticonvulsant properties.	[39]
(Ahx)_2_T^127^FIQFKKDLKEW^137^(Ahx)_2_	Improvement of coating efficiency in ELISA immunoassay procedures.	[40]

Acp, Aca = 6-aminocaproic acid; SIIIA—μ-conotoxins—cone snail venom toxin.

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
