# Peer review of "The Importance of 6-Aminohexanoic Acid as a Hydrophobic, Flexible Structural Element"

_ijms, 2021, doi:10.3390/ijms222212122_

Round 1
Reviewer 1 Report
The review well covered the application of 6-Aminohexanoic acid in the areas of drug modification, the linker of the polypeptide synthesis, and biotinylation reaction, as well as polymer biosynthesis. The manuscript is well written as a review. The reviewer has no major concerns. However, the current title “6-Aminohexanoic acid as flexible element in potential active structure” should be improved, what is the meaning of in potential active structure?
Author Response
Indeed, we agree with the reviewer, the proposed title was inadequate to the content of the manuscript and stylistically incorrect. Indeed, not all the discussed structures showed biological activity, especially since Ahx did not always improve this activity. We hope you will find the more general title satisfying.
Reviewer 2 Report
Dear Editor,
The manuscript by Markowska et al. summarizes the role of 6-Aminohexanoic acid in the structure of different bioactive molecules and other applications.
Although the need and interest for a review article on Ahx may be quite limited, the manuscript needs a severe revision and restructuring before being suitable for publication. See below:
- The title is not accurate enough, since not only active structures are addressed in the paper.
- The introduction is very short and ambiguous. It is presented as a nomenclature paragraph more than a proper introduction. For example: line 22 states “ As the amino acid has been used 21 for a long time in so many fields” I recommend listing those areas.
- I also miss a general introduction of the review article
- Synthesis and obtaining methodology
- Throughout the text different abbreviations are used indifferently, and I recommend choosing a single one for better clarity.
- Some references are missing in table 1.
- I do not understand the meaning of lines 391-394, please revise.
- Paragraph 8 is titled “Ahx and a selectivity of photodinaic therapy” but it includes a mixture of topics: radiopharmatheuticals, single-molecule microscopy, etc.
- Figure 4 is confusing. Most of the applications indicated are not PDT.
Author Response
Indeed, the interest in the Ahx review article may be quite limited due to the fact that it is a molecule that has been known for quite a long time in the field of pharmacy, but its potential in terms of new linker applications may be of particular interest. One of the authors (Markowska Agnieszka) at the same time, with this article, pays a tribute to Professor Midury-Nowaczek for her long-term cooperation. Ahx was the main topic of the Professor's interests.
The sentence in line 22 "As the amino acid has been used 21 for a long time in so many fields" has been modified. The authors meant the variety of Ahx abbreviations used in the literature, but not the range of Ahx. We apologize for the accuracy.
The method of obtaining Ahx has been introduced.
Acid abbreviation only to Ahx was unified throughout the text, leaving the original abbreviations in the structures of molecules described by other authors, including tables 1 and 2, and structures in the subsection "Antifibrinolytics". However, historically speaking, aminohexanoic acid as an antifibrinolytic is known by the abbreviation EACA.
The references in the table 1 have been corrected. All literature has been revised and reorganized according to the new layout of the work. Several new references for Ahx synthesis have been introduced.
Manuscript reorganized for Section 8 "Ahx and selectivity of photodynamic therapy". Indeed, the reviewer rightly stated that the chapter covers a mix of topics: radiopharmaceuticals, monomolecular microscopy, etc. Since all the strings described in this chapter dealt with Ahx as a linker, the chapter on radiopharmaceuticals was dropped. All data from this chapter has been transferred to the chapter "Ahx as linker". We agree with the reviewer that the use of a separate chapter dedicated to the activity of Ahx compounds, rather than structure, introduced unnecessary confusion to the manuscript. We hope the work is now more organized. Figure 4 has also been changed. Elements not belonging to the described class of compounds have been removed.
Round 2
Reviewer 2 Report
Dear Editor,
The manuscript by Markowska et al. summarizes the role of 6-Aminohexanoic acid in the structure of different bioactive molecules and other applications. Manuscript has been improved after revision but still has some issues which should be addressed. See below:
- In figure 4 points “Advantages and disadvantages of inserting 6-aminohexanoic acid to structures with potential application in an anti-cancer therapy”, are the authors sure that the Ahx linker is involved in the improvement on activity, pharmacokinetic and other properties of the conjugate or it was just a linker to joint different constructs. In my opinion, I think that the importance of Ahx is a bit exaggerated when it appears as a point of connection between different parts. I think the choice is more based on availability of the linker than to a rational design based on the importance of Ahx. Clear examples should be included showing the differences of including or not Ahx.
Author Response
I fully agree with the reviewer that Ahx is a readily available compound that easily changes only the physico-pharmacokinetic properties of the synthesized structures, so emphasizing the influence of Ahx on biological activities is exaggerated. So I completely changed figure 4, highlighting the importance of Ahx only as a linker in structures with specific peptides.
Round 3
Reviewer 2 Report
Dear Editor,
The manuscript by Markowska et al. summarizes the role of 6-Aminohexanoic acid in the structure of different bioactive molecules and other applications. Manuscript has been improved after revision and I think it is suitable for publication in present form.